# Excess Accumulation of Lipid Impairs Insulin Sensitivity in Skeletal Muscle

**DOI:** 10.3390/ijms21061949

**Published:** 2020-03-12

**Authors:** Sung Sup Park, Young-Kyo Seo

**Affiliations:** Aging Research Center, Korea Research Institute of Bioscience and Biotechnology, Daejeon 34141, Korea; sspark@kribb.re.kr

**Keywords:** insulin sensitivity, free fatty acid, skeletal muscle, metabolic dysfunction

## Abstract

Both glucose and free fatty acids (FFAs) are used as fuel sources for energy production in a living organism. Compelling evidence supports a role for excess fatty acids synthesized in intramuscular space or dietary intermediates in the regulation of skeletal muscle function. Excess FFA and lipid droplets leads to intramuscular accumulation of lipid intermediates. The resulting downregulation of the insulin signaling cascade prevents the translocation of glucose transporter to the plasma membrane and glucose uptake into skeletal muscle, leading to metabolic disorders such as type 2 diabetes. The mechanisms underlining metabolic dysfunction in skeletal muscle include accumulation of intracellular lipid derivatives from elevated plasma FFAs. This paper provides a review of the molecular mechanisms underlying insulin-related signaling pathways after excess accumulation of lipids.

## 1. Introduction

Skeletal muscle is thought to be the primary organ responsible for insulin-stimulated glucose tolerance; therefore, the resistant response of skeletal muscle to insulin stimulation is considered a key step in the progress of metabolic diseases [1]. The blood glucose levels are tightly regulated to meet the energetic demands of tissue in the body, blood glucose is easily derived from intestinal absorption during food digestion, and is partly supported by the breakdown of glycogen (glycogenolysis) and the generation of glucose from certain non-carbohydrate carbon substrates during the fasting state (gluconeogenesis) [2,3,4]. Glucose uptake stimulated by insulin leads to increased lipogenesis and glycogen and protein synthesis, and it inhibits lipolysis, glycogenolysis, and protein breakdown in skeletal muscle [5,6]. Up to 75% of insulin-dependent glucose reposition occurs in skeletal muscle. A gradual decrease in the sensitivity of skeletal muscle to insulin is considered a primary event in metabolic disease progression, such as of type 2 diabetes (T2D) [7]. The accumulation of lipids is thought to occur in skeletal muscle insulin resistance by altering the subcellular localization of diacylglycerols (DAGs) and ceramides, which accelerate peripheral inflammation and damage, progressing to heart failure, nonalcoholic fatty liver disease, obesity, renal anemia, sarcopenia, and diabetes [8]. Triacylglycerol (TG) is mostly a neutral and harmless type of lipid for storage; however, alternative conversions produce detrimental lipid intermediates.

Exercise increases skeletal muscle glucose uptake through an insulin-independent pathway [9,10], indicating that muscle contraction directly impacts glucose homeostasis. Chronic resistance exercise training improves the beneficial effects on aged skeletal muscle through increasing muscle strength and maintaining muscle mass through the activation of autophagy formation [11]. Aerobic exercise training has been shown to increase glucose transporter type 4 (GLUT4) protein expression levels by 20%–70% in human [12] and rodent [13] skeletal muscle, suggesting that aerobic training promotes acute insulin-triggered and exercise- and muscle contraction-stimulated glucose uptake in muscle.

Although molecular links between insulin resistance and T2D remain unclear, several mechanisms have clarified that the resistance in skeletal muscle involved the intracellular accumulation of lipid derivatives, such as DAG and ceramides, which are closely associated with elevated plasma free fatty acids (FFAs) [12]. The accumulation of total body fat suggests that aberrant storage of lipids or lipid intermediates in skeletal muscle contributes to the development of insulin resistance [13,14]. Here, we primarily review and discuss insulin signaling pathways and function in skeletal muscle, and how an excess of fuel source accumulation in skeletal muscle impairs insulin sensitivity.

## 2. Insulin Signaling Pathway and Glucose Uptake in Skeletal Muscle

Maintaining metabolic homeostasis involves the concerted communication efforts of multiple tissues including the digestive system, pancreas, and adipose and muscle tissues [1]. Homeostasis maintenance requires coordinating systemic energy demands among tissues that store or use energy substrates, carbohydrates, and lipids. The development of muscle insulin resistance has been observed in genetic mouse models of selective muscle insulin resistance, resulting from the muscle-specific inactivation of the insulin receptor and/or Glut4. The insulin is released from pancreatic β-cells when nutritional carbohydrates are abundant. The majority of insulin-stimulated glucose uptake occurs in skeletal muscle (Figure 1). Defects in the sensing of the energy status and the appropriate response result in metabolic diseases [15,16]. The metabolic situation of an individual, whether in fed or fasting states, determines the use of glucose as a primary source of ATP production [16].

When plasma glucose levels are elevated in a fed state, insulin secretion enhances glucose uptake by skeletal muscle, which also leads to reduced lipolysis in adipose tissue. Insulin binds to the insulin receptor, leading to activation of phosphatidylinositol 3-kinase (PI3K) and an increase in the level of intracellular phosphatidylinositol 3,4,5-trisphosphate (PIP3). PI3K binds to insulin receptor substrate (IRS) proteins, resulting in phosphorylation and activation of AKT, which translocates the GLUT4 to plasma membrane, carrying subsequent glucose uptake into the skeletal muscle. Among the four types of well-characterized glucose transporter proteins located in the plasma membrane of the cells, GLUT4 is insulin-sensitive and expressed at the cell surface in cytoplasmic vesicles in muscle and adipose tissues. In skeletal muscle, activation signals flow through tyrosine kinase on the insulin receptor (IR)/insulin receptor substrate (IRS)/PI3K/AKT2/AKT substrate 160 (AS160, also known as TBC1D4) to the GLUT4 storage vesicles [17,18]. One of the best characterized AKT substrates is TBC1D4, known as the GTPase-activating protein (GAP). TBC1D4 is a negative regulator in the insulin transduction coordination, in which deletion of TBC1D4 elevates GLUT4 levels in the plasma membrane in the absence of insulin stimulation. Active research on how TBC1D4 exerts its regulation is underway.

Although AKT phosphorylation of TBC1D1/TBC1D4 might be a key step in GLUT4 translocation, additional alterations in AKT signaling in skeletal muscle might be necessary to trigger glucose intolerance and insulin resistance [19]. Jaiswal et al. generated muscle-specific Akt-1 and -2 double knockout (KO) models (M-AktDKO) to demonstrate that chronic decrease in AKT activity alone in skeletal muscle is insufficient to perturb insulin sensitivity in mouse model. Sylow et al. reported that inhibition of AKT prevents insulin-stimulated RAC1 activation in both extensor digitorum longus (EDL) and soleus muscle; combined inhibition of both AKT and Ras-related C3 botulinum toxin substrate 1 (RAC1) produced an additive effect on the decrease in insulin-stimulated glucose uptake [20]. Takenaka et al. suggested that RAC1 is a modulated downstream molecule of AKT2 in the regulation of insulin-stimulated glucose uptake in gastrocnemius muscle [21]. 

Phosphoinositide phosphatases are reported to negatively regulate the insulin-induced signaling pathway. Heterozygous expression of SH2-domain containing Phosphatidylinositol-3,4,5-trisphosphate 5-phosphatase 2 (SHIP2) in mouse enhances GLUT4 recruitment and glycogen synthesis in skeletal muscles, leading to increased glucose tolerance and insulin sensitivity [22]. The skeletal muscle and kidney enriched inositol phosphatase (SKIP) is an inositol polyphosphate 5-phosphatase enriched in kidney and an abundantly expressed PIP3 phosphatase in the skeletal muscle. Knockdown (KD) of SKIP in C2C12 cells markedly increases glucose transport into the cytoplasm in response to insulin stimulation [23]. Ijuin et al. reported that SKIP inhibits the regulation of insulin-induced phosphatidylinositol 3-kinase signaling, and heterozygous knock-out of SKIP in mice causes increased insulin signaling, leading to enhanced glucose uptake in the skeletal muscle [24]. Therefore, SKIP is the possible regulator specific for insulin resistance induced by a high-fat diet in skeletal muscle among PIP3 phosphatases. In addition to regulation of insulin-stimulating signaling, SKIP mediates skeletal muscle differentiation, inferred to increase insulin-like growth factor (IGF) transcription and to potentiate the IGF–PI3K–AKT–mTOR auto-regulation loop during myogenesis [24].

As a major storage component for glucose, insulin resistance in skeletal muscle affects whole-body glucose metabolism. Impaired glucose transport and reduced insulin responsiveness are observed in obese human muscle fiber [25]. Impaired fasting glucose is primarily characterized by hepatic insulin resistance, whereas muscle insulin resistance easily progresses to metabolic disease. In skeletal muscle from type 2 diabetic patients, IRS-1 tyrosine phosphorylation, PI 3-kinase activity, and glucose transport activity were impaired, whereas insulin receptor tyrosine phosphorylation, MAP kinase phosphorylation, and glycogen synthase activity were normal [26,27]. Decreased insulin receptor molecules cause insulin resistance and impaired glucose tolerance, and defective insulin receptor tyrosine kinase in skeletal muscle triggers insulin resistant obesity. Muscle-specific ablation of Grb10 caused increased skeletal muscle glucose uptake in a hyperinsulinemic-euglycemic clamp study without changes in liver insulin sensitivity [28]. Reduced glucose uptake is observed in soleus muscle only under insulin-stimulated condition, and in adipocytes under both basal and stimulated conditions in TBC1D4-deficient mice, too [29].

## 3. High Fat Accumulation Triggers Insulin Resistance in Skeletal Muscle

Lipid metabolites and their mobilization are controlled mainly by the adipose tissue [30]. Lipogenesis is a main process in which FAs are produced from glucose, promoting the biosynthesis of TG and storage into lipid droplets in adipocytes [31,32]. The uptake of circulating free fatty acid (FFA) by the liver, skeletal muscle, and other tissues is a main step in lipid storage and mobilization. When the amount of FFA exceeds the absorption capacity of the adipocytes, lipids spill over into non-adipose tissues, heart, liver, and skeletal muscle. The burden of the excess amount of intracellular lipids is harmful to the cells and causes cellular dysfunction [33]. Excess lipid accumulation to DAGs and ceramides in skeletal muscle is correlated with insulin resistance in little-exercise individuals [34]. Skeletal muscle fatty acid uptake, which is regulated via LPL and fatty acid transport proteins, FATP or CD36, might be higher in conditions with impaired glucose metabolism. In obese, insulin-resistant, or T2D patients, an increased intramuscular TAG concentration has been linked with a reduced oxidative capacity, which could lead to FA storage, and thereby provide a mechanism for lipid accumulation within skeletal muscle.

Increasing levels of nuclear sterol regulatory element binding protein -1(SREBP-1), a transcription factor regulating muscle mass, induces both in vitro and in vivo muscle cell differentiation and increases in muscle mass [35]. As a clue to the role of SREBP-1 in muscle differentiation, Guillet-Deniau et al. first reported that it is expressed in various types of muscle and in primary cultures of muscle stem cells (satellite cells), demonstrating that glucose rapidly stimulates the expression and maturation of SREBP-1 as well as lipogenic gene expression in muscle satellite cells [36,37]. SREBP-1 mRNA is observed in adult rat skeletal muscles and this expression decreases in diabetic condition [38]. Glucose induces de novo lipogenesis in rat muscle satellite cells through the SREBP-1c-dependent pathway through which glucose stimulates the expression and maturation of SREBP-1c more rapidly than insulin, leading to an increased lipogenic flux and intracellular lipid accumulation [36,39]. Lecomte et al., provide the evidence that the SREBP-1 proteins may regulate the hypertrophic effects of growth factors not only negatively through induction of the BHLHB2 and BHLHB3 repressors but also positively through the control of PI3K/PKB signaling pathway [38]. Further investigations are required to unveil the impact of SREBP-1 on signaling pathways in skeletal muscle cells. The SREBP-1-mediated effects on BHLHB2 and BHLHB3 activity thus defines a novel negative regulation pathway in skeletal muscle cell development inferring that SREBP-1 leads to atrophy with a decrease in muscle atrophic factors [38]. In SREBP-1 overexpressing cells, a combined decrease in the expression of the myogenic regulatory factors MYOD1, MYOG and MEF2C previously was reported. The myogenic regulatory factors may be required to maintain muscle homeostasis, and may play a role in muscle plasticity in response to both hypertrophic (i.e., exercise) and atrophic (i.e., denervation) stimuli [40].

Conversely, KD of SREBP-1 by small interfering RNAs (siRNAs) totally abolished the glucose-induced upregulation of lipogenic enzymes, suggesting that glucose-induced SREBP-1c expression and maturation could be implicated in muscle lipotoxicity. During de novo lipogenesis, the conversion of glucose to FAs includes a coordinated series of enzymatic reactions, of which fatty acid synthase (FAS) is the key rate-limiting enzyme that regulates the conversion of malonyl-coenzyme A (CoA) into palmitate, which is thereafter converted into complex fatty acids [33]. Analysis of microarray data from muscle cells overexpressing SREBF1 suggests a role in the regulation of myogenesis [41]. Overexpression of SREBP-1 leads to atrophy of both differentiated myotubes in vitro and tibialis muscle in vivo, indicating that increased nuclear SREBP-1 content induces myotube atrophy, with a decrease in muscle atrophic factors and sarcomeric proteins, FBXO32 (atrogin1), MURF1 (MuRF-1/TRIM63), and FOXO1. SREBP-1 expression is increased through the AKT/mTOR pathway and participates in the control of lipid and cholesterol metabolism associated with the regulation of cell size [40]. Due to the major roles of insulin, growth factors, and the PI3K/AKT signaling pathway on muscle development and growth, SREBP-1 proteins may thus regulate the hypertrophic effects of growth factors. Further studies are required to elucidate the effects of SREBP-1 on signaling pathways in skeletal muscle.

Of the fuel sources for energy production in skeletal muscle, lipid is available from intramuscular TG droplets within the muscle fiber as well as from plasma FFAs from lipolysis. The number of bonds in the hydrocarbon chains of FFAs determines whether unsaturated or saturated fatty acid form. 4-phenyl butyric acid and tauroursodeoxycholic acid reduce endoplasmic reticulum (ER) stress in cells and tissues, and promote normalization of hyperglycemia, restoration of systemic insulin sensitivity and fatty liver disease, and enhancement of insulin action in liver, muscle, and adipose tissues in obese and diabetic model mice [42]. In humans, treatment with tauroursodeoxycholic acid increases hepatic and muscle insulin sensitivity by ~30% and phosphorylation of IRS and AKT, without effects on ER stress in muscle or adipose tissue [43].

Recent research of insulin receptor signaling indicates that the accumulation of FFA in muscle can interfere with insulin signaling and produce insulin resistance. Several mechanisms involved in the development of insulin resistance in skeletal muscle have been proposed. One of the substantial factors is the intracellular accumulation of lipid derivatives, such as DAG and ceramides, which is a result of aberrantly elevated FFAs in circulating plasma [44,45].

Ahn et al. recently demonstrated that the enriched transcription factor MondoA induces upregulation of many genes connected with fatty acid metabolism, the hexosamine biosynthetic pathway, glycogen synthesis, and insulin signaling in human skeletal muscle [46]. MondoA depletion negatively regulates the expressions of thioredoxin interacting protein (TXNIP) and the arrestin domain containing 4 (ARRDC4), which act as suppressors of insulin signaling. KD of MondoA downregulated the PKC-θ transcript encoding a negative modulator of insulin signaling as well. Mice with muscle-specific MondoA deficiency are partially protected from insulin resistance and muscle TG accumulation in the context of diet-induced obesity. The expressions of TG species containing acyl groups C16:1, C18:0, and C18:1 are suppressed in MondoA-deficient muscle [47,48]. These results are consistent with a role for MondoA in the diversion of fatty acids into muscle TG stores in the context of caloric excess in muscle in vivo.

Aging is associated with many metabolic changes, including lipid accumulation and the development of insulin resistance. Skeletal muscle mass and strength decrease with aging (sarcopenia); type II fibers rather than type I fibers are selectively lost [49]. Type I fibers are adapted to manage excess FA, and type II fibers are more vulnerable to lipid stress. Muscle fiber types have different insulin sensitivities. Type I fibers have much higher protein contents of IR, GLUT4, hexokinase II (HK2), and glycogen synthase (GS), and lower protein expression levels of AKT2, TBC1D1, and TBC1D4 [50]. Many biological aging processes do not occur independently and involve interactions between multiple tissues. The results of lipid accumulation during aging are heterogenous between fiber types. Excess lipid in skeletal muscle changes the phospholipid composition and then the functional integrity of the sarcoplasmic reticulum (SR) [51]. KD expression of choline/ethanolamine phosphotransferase 1 increases muscle insulin sensitivity in high-fat diet mice and decreases SR Ca^2+^ ATPase-dependent calcium uptake, activating calcium signaling and increasing insulin sensitivity [51]. Lipid accumulation triggers oxidative stress. Oxidative stress induces many deleterious effects: alteration of redox balance, lipid peroxidation, ER stress, protein mis-folding, decrease in protein synthesis, activation of apoptosis, and disturbance of calcium flux. Oxidative stress in skeletal muscle induced by a high-fat diet is different between fiber types [52]. Highly oxidative soleus muscle emits one-fifth and one-third of the hydrogen peroxide obtained from extensor digitorum longus and epitrochlearis muscle, respectively. These data suggest that oxidative stress is intrinsically fiber-type-specific, which is harmful under the conditions of excess lipid accumulation.

As mentioned above, a positive role of PKCδ is considered in insulin-stimulated glucose uptake in skeletal muscle cells. Muscle PKCδ levels increase with age in mice, and muscle-specific deletion of PKCδ improves muscle insulin resistance and whole-body insulin sensitivity in aged mice [53]. Fatty-acid-binding protein (FABP), fatty acid transport protein 1 (FATP1), and cluster of differentiation 36 (CD36) are widely expressed plasma lipid transporters in rodent and human skeletal muscle plasma membrane [54,55]. By controlling entry of long-chain fatty acids (LCFAs) across the barrier of the inner mitochondrial membrane for subsequent β-oxidation, the carnitine shuttle system could influence skeletal muscle substrate switch. One of the well-investigated components of this system is malonyl-CoA sensitive carnitine palmitoyltransferase 1 (CPT1), which is expressed on the mitochondrial outer membrane, as summarized in Figure 2.

## 4. Maintaining Cell Homeostasis in Skeletal Muscle

Dietary and hepatic TG (approximately 100 g/day) are hydrolyzed to free fatty acids and monoacylglycerol by intestinal lipases. VLDL particles are transported to peripheral tissues where the TGs are hydrolyzed by membrane-bound lipoprotein lipase, and fatty acids are released by transport proteins and partially by diffusion [56,57]. TG levels within the skeletal muscle, or intramuscular triglyceride (IMTG), exist in healthy muscles in small quantities, whereas they are increased in obese individuals stored in lipid droplets (LDs). IMTG levels within the skeletal muscle are an important energy source for muscle contraction, which are gradually increased in athletes and with exercise [58,59] However, increased IMTG is closely related with insulin resistance in obese and diabetic patients. Therefore, understanding the process of regulating IMTG dynamics in muscle differentiation and growth is important for treating muscle loss. IMTG degradation leads to abnormal accumulation of DAG and ceramide, resulting in excessive activity of atypical PKC, and inhibiting insulin signaling to induce insulin resistance [60]. The mechanism underlining DAG-induced IR is that increased plasma FA concentration leads to accumulation of intramyocellular acyl-CoAs and DAG, which activates isoforms of protein kinase C (PKC). In turn, the activities of PI3K and IRS in the insulin signaling pathway for muscle glucose uptake are decreased [61,62]. Three groups of fatty acid transporters have been identified: FABP, CD36 (also known as FAT, SCARB3, GP88, and GPIIIB), and FATP [63]. Of these, CD36 is the best-studied fatty acid carrier in skeletal muscle. It is located within the cytosol of myocytes and acts as a shuttle between plasma membrane and cytoplasm to carry FFA from plasma into muscle cells as GLUT4 in the glucose uptake process [61,64].

The use of fatty acids generated by IMTG degradation occurs in the mitochondria. When mitochondrial dysfunction occurs due to various causes, fatty acid oxidation induces lipotoxicity in muscle cells, leading to insulin resistance, decreased glucose used, and causing metabolic disease due to energy imbalance [65,66]. When mitochondrial oxidative capacity is impaired, metabolic substrate catabolism decreases and then fatty acids availability increases in intracellular myocytes, resulting in channeling toward lipotoxic DGA, which has been associated with insulin resistance. Increased nutrient inputs also trigger an increase in reactive oxygen species (ROS) production in mitochondria. Increased mitochondrial ROS elicits oxidative damage to mitochondrial DNA, proteins, and lipids, perturbs insulin sensitivity, and can promote the removal of damaged mitochondria by mitophagy [66].

Recently, research has focused on the regulation of mitochondrial autophagy, called mitophagy, which is essential for maintaining cell homeostasis as a process regulating the life cycle of all mitochondria. In skeletal muscle, damaged mitochondria are targeted for degradation via either PTEN-induced kinase 1 (PINK1)/ PARKIN-mediated mitophagy pathways, ultimately leading to autophagosome engulfment, increasing pancreatic β cell insulin secretion, and improving muscle insulin resistance [67,68]. At least two mitophagy pathways have been characterized. In the first, PINK1 accumulates on the mitochondrial outer membrane, leading to its phosphorylation and activation [69,70], then recruits and activates PARKIN, which subsequently ubiquitinates outer membrane proteins in the mitochondria and attracts the adapter protein SQSTM1/p62 to the ubiquitin chains [71]. p62 binds to the autophagosome embedded protein microtubule-associated protein 1A/1B-light chain 3 (LC3)-II, which was lipidated through a series of reactions from the inactive precursor LC3-I. Subsequently, the cargo is engulfed in a double membrane autophagosome and transported along microtubules to a lysosome, where its contents are degraded. The second targeting mechanism includes the recruitment of the receptors BCL2-interacting protein 3 (BNIP3) and BNIP3-like protein (NIX) to the mitochondria [72], which are capable of binding to LC3-II, leading to mitophagy.

Another mitophagy mediator, FUN14 domain containing 1 (FUNDC1), has been reported to be associated with muscle fat use and motor capacity [73]. FUNDC1 plays a critical role in controlling muscle mitochondrial quality as well as metabolic homeostasis. Ablation of FUNDC1 in skeletal muscle resulted in LC3-mediated mitophagy defects, leading to impaired mitochondrial energetics. Mice lacking FUNDC1 in muscle were protected against chronic high-fat-diet-induced obesity and insulin resistance, despite reduced muscle mitochondrial energetics. These results strongly suggest that muscle mitophagy pathways could potentially be targeted to counteract metabolic disorders, such as obesity.

## 5. Exercise Effects on Glucose Uptake in Muscle Insulin Resistance

Insulin-stimulated glucose uptake is increased by exercise in skeletal muscle rather than in adipose tissue [6,74,75]. Physical exercise induces glucose uptake insulin dependently or independently in skeletal muscle without insulin receptor signaling by inducing translocation of GLUT4 to the cell surface [76]. This effect continues for several hours after exercise with increasing in insulin sensitivity, which leads to insulin-dependent glucose transport becoming prominent [76,77]. Exercise enhances insulin-induced phosphorylation of TBC1D4 and improves insulin sensitivity in diabetes [78]. Protein phosphatase 1-α (pp1-α) and abundant serine/threonine protein phosphatases expressed in skeletal muscle regulate TBC1D4 dephosphorylation [79]. Acute exercise reduces the phosphorylation rate of TBC1D4 concomitant with increased glucose uptake in insulin-induced rat skeletal muscle [80].

Depending on the cell types, activation of autophagy in insulin-responsive tissues such as muscle, liver, and adipose tissues improves insulin response and function. Yamamoto et al. reported that in response to a high-fat-diet challenge, Beclin-1(Becn1)-mediated autophagic hyperactivation increases insulin sensitivity by reducing ER stress [81]. This observation unveiled roles of autophagy in metabolic regulation in insulin-responsive cells versus insulin-secreting β cells. Besides the enhancement of GLUT4, the exercise-induced activation of AMPK mediates autophagy activation, which may contribute to beneficial metabolic effects on glucose uptake in skeletal muscle. Voluntary wheel-running exercises for five weeks showed significant increases in basal autophagy flux, expressions of autophagy-related proteins, mitochondrial biogenesis, capillary density, and endurance capacity in skeletal muscles of wild type mice compared to autophagy-deficient mice [82]. The contribution of high-intensity interval training compared with continuous moderate-intensity training has been related to improved autophagic adaptation in rat skeletal muscles. These observations strongly suggest that exercise-induced enhancement of basal autophagy activation might play an essential role in skeletal muscle performance and insulin sensitivity. Long-term resistance exercise training increases maximal capacity carrying lead weights through p62-mediated autophagy activity in chloroquine-induced rat myopathy [83]. The results indicate the therapeutic possibility that chronic resistance exercise training contributes to the improvement of muscle-related disease conditions such as muscle weakness and loss and insulin-related dysregulation of glucose by promoting autophagy activity.

Endurance exercise increases oxidative capacity and insulin sensitivity, even though lipid content is elevated in skeletal muscle [84]. These results were reported after a 12-week combined endurance-resistance exercise training program in subjects with similar total intramyocellular lipid contents but different insulin sensitivities. Combined exercise increased insulin sensitivity in both control and prediabetic groups, but significantly decreased lipid droplet volume in the subsarcolemmal space despite no differences in intermyofibrillar lipid droplet volume [85]. Lipid droplets in the subsarcolemmal space seem to be more sensitive to exercise training than those in intermyofibrillar region of skeletal muscle. Insulin sensitivity associated with exercise is attributed to spatial distribution of lipid droplet skeletal muscle myocytes. Daemen et al. showed that combined exercise-trained subjects are insulin sensitive, and their high levels of small lipid droplets accumulate in the intermyofibrillar space of oxidative type I muscle fibers, whereas T2D patients are insulin resistant and possess large lipid droplets in the subsarcolemmal region of type II muscle fibers [86].

Exercise-dependent regulation of GLUT4 is posttranscriptional and mTOR-independent in skeletal muscle [59]. AMPK activation increases skeletal muscle insulin sensitivity in obese mice [87,88], and AMPK-KO mice did not exhibit exercise-dependent muscle insulin sensitivity, which is associated with phosphorylation of TBC1D4 without insulin signaling such as phosphorylation of AKT [89]. Rac1 is involved in glucose uptake via regulation of GLUT4 translocation in mouse muscle, which is independent from exercise-induced AMPK activation [16]. However, conventional protein kinase Cα (cPKCα), predominantly expressed isoform in skeletal muscle, does not contribute to exercise-induced glucose uptake because glucose uptake in cPKCα-KO mice was not changed compared with the wild type [45]. Insulin causes association of hexokinase II with the mitochondrial outer membrane, which facilitates phosphorylation of glucose near the location of energy production [90]. Exercise increases hexokinase II transcription but not activity in posttranscriptional defective insulin resistant states [91] and activates glycogen synthase via dephosphorylation, enhancing glucose storage in skeletal muscle [92]. In obese men, acute exercise enhances glucose uptake via increased oxidative stress [93]. Exercise-induced glucose uptake in skeletal muscle is exhibited in Figure 3 for skeletal muscle tissue growth.

Autophagy formation related to lipid metabolism (lipophagy) was described in skeletal muscle. Exercise induces small-diameter lipid droplet formation in subsarcolemmal space rather than in the intermyofibrillar regions of skeletal muscle [86]. Artificial increase in mitochondrial contents by muscle-specific overexpression of PGC-1α without exercise plays a role in the initial development of glucose intolerance induced by high-fat diet feeding and in the activation of autophagy formation, but does not contribute to exercise adaptation, whereas the physical activity of voluntary wheel running increases wheel distance run and improves glucose tolerance induced by lipid overload, which is associated with a greater LCII/LC3I ratio of autophagosome regardless of PGC-1α content [94]. The liver, a central organ for the regulation of lipid metabolism, reponds differently to exercise. Chronic exercise after a high-fat diet decreases hepatic lipid droplet size and liver damage along with reducing fasting glycaemia, insulin, and total cholesterol and triacylglyceride plasma levels without hepatic declines or changes in triacylglycerides levels, lipid droplet numbers, and autophagy markers [95].

## 6. Conclusions

Understanding of the mechanisms that regulate metabolic homeostasis in skeletal muscle has progressed in the last few decades. Glucose and FFAs are the substrates that respond to insulin signaling in the skeletal muscle. The maintenance of both glucose and FFA balances depends on dynamic interactions, showing tissue sensitivity to insulin secretion and insulin signaling pathway through muscle and adipose tissue. Modulation of PI3K/AKT and exercise-related AMPK signaling appears to be the driving mechanism associated with downstream effects on the movement of GLUT4 to the plasma membrane and fusion, and glucose uptake into skeletal muscle. TBC1D4 for AKT substrate to GLUT activation as well as SKIP appear to be the necessary regulators in the insulin transduction coordination. Glucose induces de novo lipogenesis in muscle through SREBP-1c-dependent pathway, resulting in an increased lipogenic flux and intracellular lipid accumulation. Further studies are necessary to elucidate the age-associated imbalance leading to skeletal muscle loss or the sarcopenia response to insulin sensitivity. Exercise significantly benefits insulin sensitivity through various pathways, including increase in hexokinase II transcription and dephosphorylation of TBC1D4. Endurance exercise increases oxidative capacity and insulin sensitivity, even though lipid content is elevated in skeletal muscle. Insulin sensitivity associated with exercise is attributed to spatial distribution of lipid droplets in skeletal muscle myocytes. Collectively, the results obtained from these studies may lead to the development of novel therapeutic strategies to improve regenerative capacity following aging-dependent disease. Skeletal muscle needs an efficient process to meet the energy demands of the body.

## Figures and Tables

**Figure 1 ijms-21-01949-f001:**
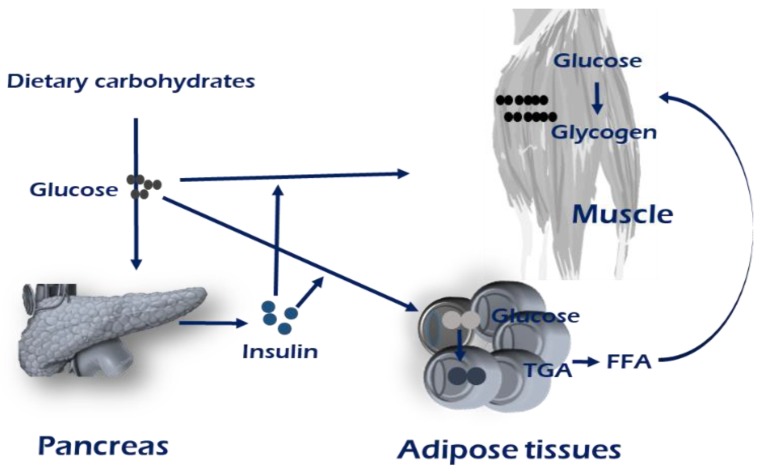
Brief overview of maintenance of energy homeostasis. Insulin stimulates the conversion of glucose into glycogen or triglycerides by increasing glucose uptake in muscle and adipose tissue. The majority of insulin-stimulated glucose uptake occurs in skeletal muscle. In obese, insulin-resistant, or T2D patients, an increased intramuscular TAG concentration has been linked with a reduced oxidative capacity, which could lead to fatty acid (FA) storage, and thereby provide a mechanism for lipid accumulation within skeletal muscle. Mechanisms regarding PI3K-AKT-mTOR signaling are mentioned in main text and the Figure 2.

**Figure 2 ijms-21-01949-f002:**
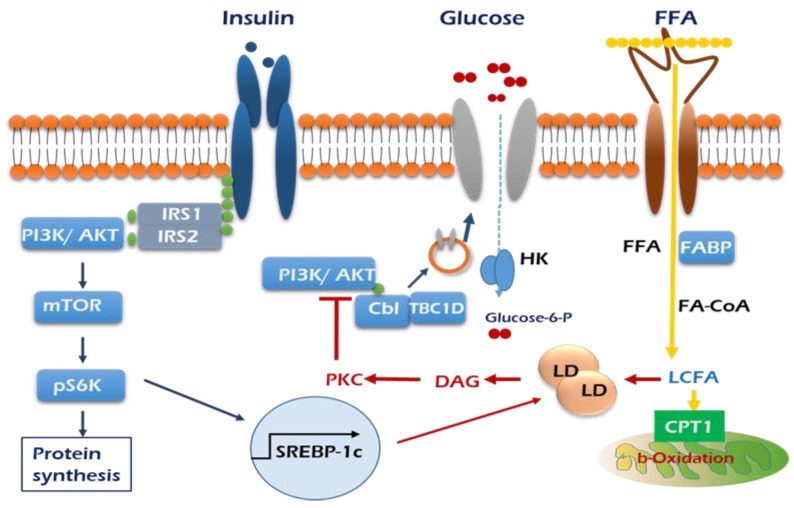
Graphic summary of insulin signaling pathways mediated by influx of lipid in the skeletal muscle. Insulin stimulation of glucose uptake occurs through translocation of GLUT4 to the plasma membrane. The resultant increase in intracellular glucose-6-phosphate production enables net glycogen synthesis. Excess plasma FFAs lead to the intramuscular accumulation of long-chain fatty acid (LCFA)-acyl CoAs. LCFA-acyl CoAs are incompletely oxidized by mitochondria and then form large lipid droplets and activate PKC, which stimulates serine phosphorylation of PI3K/AKT (red arrows), which in turn prevents the translocation of GLUT 4 to the plasma membrane and glucose uptake into skeletal muscle. 
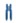
 Insulin receptor, 
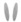
 GLUT4, 
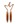
 CD36.

**Figure 3 ijms-21-01949-f003:**
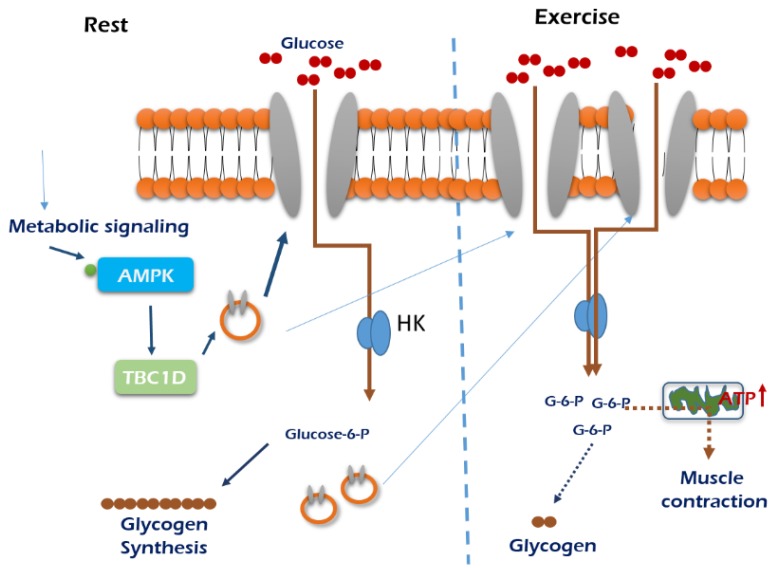
An exercise-stimulated glucose uptake in skeletal muscle. Glucose demands are minimal when muscle is in the resting stage, so glucose delivery is low. When exercising, muscle preferentially uses glycogen to supply glucose for the generation of ATP to meet increased metabolic demands. Skeletal muscle glucose uptake during exercise is mainly regulated from flux though intracellular metabolism. Under normal submaximal exercise conditions, skeletal muscle glucose uptake appears to be limited by glucose transport through the cell membrane. The mechanism stimulates GLUT4 translocation and glucose uptake appear to arise from activation of AMPK.
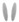
, GLUT4; HK, hexokinase.

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
