# Peer review of "Excess Accumulation of Lipid Impairs Insulin Sensitivity in Skeletal Muscle"

_ijms, 2020, doi:10.3390/ijms21061949_

Round 1
Reviewer 1 Report
Minor revision
Line 49 and others: when the acronym you are referring to is already used before, you have not to mention again the extended form; please use only the acronym
Line 76: correct vescicles
Legend to Figure1: correct line 128, PI3K: and line 129: Fig. 2; please, check the sentence as it appears incomplete.
Line 138: DAGs
Line 375-377: Skeletal muscle glucose uptake during exercise is regulated through flux through intracellular metabolism. Under normal submaximal exercise conditions, skeletal muscle glucose uptake appears to be limited by glucose transport though the cell membrane.. please correct
Author Response
Dear Reviewer 1,
Thank you very much for letting us notice the type errors and the comments. We corrected your direction point-by-point as below. These are changed in main text.
Line 49 and others: when the acronym you are referring to is already used before, you have not to mention again the extended form; please use only the acronym
Corrected as indicated in main text.
Line 76: correct vesicles
Yes, changed
Legend to Figure1: correct line 128, PI3k: and line 129: Fig. 2; please, check the sentence as it appears incomplete.
PI3k-> PI3K
Fig.2-> Figure 2
Yes, all changed
Line 138: DAGs
Yes, changed
Line 375-377: Skeletal muscle glucose uptake during exercise is mainly regulated from flux through intracellular metabolism. Under normal submaximal exercise conditions, skeletal muscle glucose uptake appears to be limited by glucose transport through the cell membrane.. please correct
Reviewer 2 Report
The authors addressed the referee's concerns appropriately.
Author Response
Thank you for your valuable time.
This manuscript is a resubmission of an earlier submission. The following is a list of the peer review reports and author responses from that submission.
Round 1
Reviewer 1 Report
Comments and Suggestions for Authors
This review shows how an excess of lipid accumulation, derived from elevated plasmatic levels of free fatty acids, impairs insulin sensitivity in skeletal muscle leading to metabolic disorders. It deals also with the molecular mechanisms underlining altered insulin signaling pathways.
The paper is well organized, but some minor aspects would be considered to clarify some details just to make it more fluent to read.
Specific recommendations for revision
Abstract
Line 9: The acronym FFA generally is used as a plural. Thus it would be better to use free fatty acids (FFA or FFAs) and not free fatty acid. The same acronym has to be used in all the text and in the figures with the aim to armonize it.
Section 1, Introduction
Line 24: please remove the word Because; the sentence has to be modified as below: The blood glucose….in the body;
Line 41: please change autophagy with autophagosome formation or remove the word formation.
Line 42: please, insert the expanded form of GLUT4 (it is the first citation).
Line 42 and 43: please insert the bibliographic reference(s) regarding human and rodent skeletal muscle.
Line 47: please, insert the acronym DAG in the brackets; armonize then all diacylglicerol of the text (lines 128, 167, 242)
Line 48: please check the acronym FFA or FFAs (as indicated previously in the abstract).
Line 51: please change the sentence as below: …in skeletal muscle and how an excess of fuel source accumulation….
Section 2, Insulin signaling pathway and glucose uptake in skeletal muscle
Line 59: please, insert the bibliographic reference after the word gene.
Line 62: please wrap lines 63 and 64 after the brackets. The line 65 has to be consecutive.
Line 72: please insert the word in before cytoplasmatic.
Line 75: please change vesicle in vesicles.
Line 80: TBC1D1 is first nominated; before it is mentioned only TBC1D4. It would be better to introduce that they are members of TBC1 domain family, with differential expression pattern throughout insulin-responsive target tissues and introduce TBC1D1 also before.
Line 85: please, insert the expanded form of RAC1 (it is the first citation).
Line 90: pay attention to the verb tenses; it would be better change were with are.
Line 91: please, insert the expanded form of SHIP2 (it is the first citation).
Line 93: SKIP is the acronym of skeletal muscle and kidney enriched inositol phosphatase; please, it would be better to cite first the extended form and the acronym in brackets.
Line 94: please, insert the acronym KD in the brackets; armonize then all the knockdown of the text (lines 139, 173, 188).
Line 102: please insert the bibliographic reference after the word myogenesis.
Line 103: the sentences does sound no good; it would better to modify it in order to be more clear.
Line 105: please correct fiber with fibers.
Line 107-109: the sentences does sound no good; it would better to modify it or to insert some punctuation in order to result more clear.
Line 113-114: pay attention to the repetition of the word observed; it seems that the subject of the sentence is lacking after the comma.
Section 3, High Fat Accumulation Triggers Insulin Resistance in Skeletal Muscle
Line 121: please, insert the before adipose tissue.
Line 122: please, insert the acronym (it is the first citation); pay attention to cite FA or FAs accordingly to the acronym that you have chosen (FFA or FFAs)
Line 124: please, delete free fatty acid; use only the acronym.
Line 129: please, insert the bibliographic reference after the word individuals. Wrap line 129 after the word individuals. Please, insert the extended form of SREBP-1.
Line 130: delete in before the word muscle.
Lines 136-138: the sentences does sound no good; it would better to modify it. Here you cite SREBP-1-c and not SREBP-1 and after (line 139) again SREBP-1. It would be better to armonize.
Lines 144,158,170,178, 233, 237, 241: Delete fatty acids and insert the acronym FA or FAs.
Line 145: please, correct SREBP1.
Line 149: please insert the bibliographic reference after the word FOXO1.
Line 150: PKB: it would be better to cite AKT, already cited before.
Line 159: animals?
Line 172: please, insert the expanded form of TXNIP and ARRDC4 (it is the first citation).
Lines 173 and 174: please, pay attention to the verb tenses.
Line 184: please insert the expanded form of GS (it is the first citation) and the acronym of hexochinase (as in the figure 3).
Line 198: the authors cite as mentioned above, regarding PKCδ, but it is not discussed before.
Line 203: please insert the acronym (LCFA as indicated in the Figure 2).
Section 4, Maintaining Cell Homeostasis in Skeletal Muscle
Line 246: insert a reference.
Line 250: please insert the expanded form of PINK (or PINK1, see line 252)/PARKIN (it is the first citation).
Line 255: please insert the expanded form of SQSTM1 also known as ubiquitin-binding protein p62 (p62).
Line 262: please, insert the expanded form of FUNDC1.
Lines 262-264: the sentence has to be modified as it contains a repetition.
Section 5, Exercise Effects on Glucose Uptake in Muscle Insulin Resistance
Line 275: change increase with increasing
Line 280: please change with increased glucose uptake in insulin-induced rat skeletal muscle with increased insulin-induced glucose uptake in rat skeletal muscle.
Line 284: please, insert the expanded form of Becn1.
Lines 321, 323, 359: please insert the acronym HK.
Lines 338 and 339: pay attention and harmonize tryacylglycerides.
Line 362: change droplet with droplets and add in before the word skeletal.
Figure 2: it would be better to insert the extended form of the main acronyms.
It would help a lot to have at the end of the manuscript a list with all the acronyms used in the text and the extended forms of the names.
References:
Correct the ref 31, 41, 42.
Author Response
Response to Reviewer
In this review, we discussed mainly how excess FFAs and lipid accumulation in skeletal muscle impair insulin sensitivity and cellular homeostasis as well as informed the molecular mechanisms underlining altered insulin signaling pathways. And we tried to summarize the main mechanisms leading to the development of insulin resistance in skeletal muscle. Skeletal muscle is responsible for the body’s energy expenditure, participating in glucose and lipid uptake, and other metabolic processes. But, an association of declining skeletal muscle mass on insulin resistance has not been fully investigated, yet. We hope that this report may be very informative to understand current consequences of metabolic responsibility of skeletal muscle. We really appreciate reviewer for bunch of comments and directions. We corrected reviewer’s direction point-by-point as below and main manuscript, too.
Comments and Suggestions for Authors
This review shows how an excess of lipid accumulation, derived from elevated plasmatic levels of free fatty acids, impairs insulin sensitivity in skeletal muscle leading to metabolic disorders. It deals also with the molecular mechanisms underlining altered insulin signaling pathways.
The paper is well organized, but some minor aspects would be considered to clarify some details just to make it more fluent to read.
Specific recommendations for revision
Abstract
Line 9: The acronym FFA generally is used as a plural. Thus it would be better to use free fatty acids (FFA or FFAs) and not free fatty acid. The same acronym has to be used in all the text and in the figures with the aim to armonize it.
Yes, all FFA changed to FFAs through all the text
Section 1, Introduction
Line 24: please remove the word Because; the sentence has to be modified as below: The blood glucose….in the body;
Yes, changed as suggested
Line 41: please change autophagy with autophagosome formation or remove the word formation.
Yes, changed as suggested
Line 42: please, insert the expanded form of GLUT4 (it is the first citation).
Yes, changed as suggested
Line 42 and 43: please insert the bibliographic reference(s) regarding human and rodent skeletal muscle.
Yes, changed as suggested
Line 47: please, insert the acronym DAG in the brackets; armonize then all diacylglicerol of the text (lines 128, 167, 242)
Yes, changed all
Line 48: please check the acronym FFA or FFAs (as indicated previously in the abstract). Yes, changed
Line 51: please change the sentence as below: …in skeletal muscle and how an excess of fuel source accumulation….
Yes, changed
Section 2, Insulin signaling pathway and glucose uptake in skeletal muscle
Line 59: please, insert the bibliographic reference after the word gene.
Yes, inserted
Line 62: please wrap lines 63 and 64 after the brackets. The line 65 has to be consecutive. Yes, we did
Line 72: please insert the word in before cytoplasmatic. Yes, changed
Line 75: please change vesicle in vesicles. Yes, changed
Line 80: TBC1D1 is first nominated; before it is mentioned only TBC1D4. It would be better to introduce that they are members of TBC1 domain family, with differential expression pattern throughout insulin-responsive target tissues and introduce TBC1D1 also before.
Line 85: please, insert the expanded form of RAC1 (it is the first citation).
Yes, changed
Line 90: pay attention to the verb tenses; it would be better change were with are.
Yes, changed
Line 91: please, insert the expanded form of SHIP2 (it is the first citation).
Yes, changed
Line 93: SKIP is the acronym of skeletal muscle and kidney enriched inositol phosphatase; please, it would be better to cite first the extended form and the acronym in brackets. Yes, changed
Line 94: please, insert the acronym KD in the brackets; armonize then all the knockdown of the text (lines 139, 173, 188).
Yes, we did
Line 102: please insert the bibliographic reference after the word myogenesis.
Line 103: the sentences does sound no good; it would better to modify it in order to be more clear. Yes, changed
Line 105: please correct fiber with fibers. Yes, changed
Line 107-109: the sentences does sound no good; it would better to modify it or to insert some punctuation in order to result more clear.
Yes, we modified line 108 as ‘In skeletal muscle from type 2 diabetic patients, IRS-1 tyrosine phosphorylation, PI 3-kinase activity, and glucose transport activity were impaired, whereas insulin receptor tyrosine phosphorylation, MAP kinase phosphorylation, and glycogen synthase activity were normal.
Line 113-114: pay attention to the repetition of the word observed; it seems that the subject of the sentence is lacking after the comma.
Yes, changed ‘but is observed’ to ‘and’.
Section 3, High Fat Accumulation Triggers Insulin Resistance in Skeletal Muscle
Line 121: please, inse rt the before adipose tissue.
Yes, inserted.
Line 122: please, insert the acronym (it is the first citation); pay attention to cite FA or FAs accordingly to the acronym that you have chosen (FFA or FFAs)
Yes, changed
Line 124: please, delete free fatty acid; use only the acronym.
Yes, changed
Line 129: please, insert the bibliographic reference after the word individuals.
Wrap line 129 after the word individuals. Yes, changed Please, insert the extended form of SREBP-1.
Yes, changed
Line 130: delete in before the word muscle. Yes, changed
Lines 136-138: the sentences does sound no good; it would better to modify it. Here you cite SREBP-1-c and not SREBP-1 and after (line 139) again SREBP-1. It would be better to armonize.
Lines 144,158,170,178, 233, 237, 241: Delete fatty acids and insert the acronym FA or FAs. Yes, changed
Line 145: please, correct SREBP1. Yes, changed
Line 149: please insert the bibliographic reference after the word FOXO1.
Yes, changed
Line 150: PKB: it would be better to cite AKT, already cited before.
Yes, changed
Line 159: animals? Yes, changed to tissues
Line 172: please, insert the expanded form of TXNIP and ARRDC4 (it is the first citation). Yes, changed
Lines 173 and 174: please, pay attention to the verb tenses. Yes, changed
Line 184: please insert the expanded form of GS (it is the first citation) and the acronym of hexochinase (as in the figure 3).
Yes, changed
Line 198: the authors cite as mentioned above, regarding PKCδ, but it is not discussed before. Yes, corrected
Line 203: please insert the acronym (LCFA as indicated in the Figure 2).
Yes, changed
Section 4, Maintaining Cell Homeostasis in Skeletal Muscle
Line 246: insert a reference.
Line 250: please insert the expanded form of PINK (or PINK1, see line 252)/PARKIN (it is the first citation).
Line 255: please insert the expanded form of SQSTM1 also known as ubiquitin-binding protein p62 (p62). Yes, changed.
Line 262: please, insert the expanded form of FUNDC1. Yes changed
Lines 262-264: the sentence has to be modified as it contains a repetition.
Yes, changed
Section 5, Exercise Effects on Glucose Uptake in Muscle Insulin Resistance
Line 275: change increase with increasing
Yes, changed
Line 280: please change with increased glucose uptake in insulin-induced rat skeletal muscle with increased insulin-induced glucose uptake in rat skeletal muscle. Yes, changed
Line 284: please, insert the expanded form of Becn1. Yes, changed
Lines 321, 323, 359: please insert the acronym HK. Yes, changed
Lines 338 and 339: pay attention and harmonize tryacylglycerides. Yes, changed
Line 362: change droplet with droplets and add in before the word skeletal.
Yes, changed
Figure 2: it would be better to insert the extended form of the main acronyms.
It would help a lot to have at the end of the manuscript a list with all the acronyms used in the text and the extended forms of the names.
References:
Correct the ref 31, 41, 42.
Yes, corrected all

Reviewer 2 Report
The review manuscript entitled “Excess Accumulation of Lipid Impairs Insulin Sensitivity in Skeletal Muscle” introduces some recent progress in the area of insulin sensitivity in skeletal muscle. Lipid accumulation and insulin sensitivity are two important and hot topics in the study of metabolic disorders. New insights from these two topics will potentially benefit the treatment of diseases, such as type 2 diabetes.
Unfortunately, the manuscript has some critical issues.
1) From the title, some important questions are expected to be addressed: a) How does excess accumulation of lipids develop in skeletal muscle? Where are the lipids from? What could be the molecular mechanisms behind this process? b) How accumulation of lipids changes insulin sensitivity and what could be the corresponding signaling? c) How could exercise or other interventions affect or restore insulin sensitivity targeting lipid accumulation, and what could be the mechanisms?
For these important questions, the manuscript doesn’t provide clear and detailed answers or thoughts.
2) The authors don’t provide full names for a lot of abbreviations. For example, it is hard to believe that the full names of GLUT4 and SREBP-1 are not included in the manuscript. This demonstrates that the authors paid little attention to the quality of the manuscript.
3) All the figures are not the summaries of corresponding parts. Figure 1 and 3 are general and simple figures without providing valuable information and summaries. For figure 2, it is not clear how the authors came up with this figure, because there is no description or introduction for the interactions of different components in the figure in the manuscript. Besides, it is difficult to understand how insulin resistance develops from figure 2. Basically, the authors provide some results from current research, but cannot provide appropriate summaries and new thoughts or ideas.
4) References are needed for the contents from line 240 to line 246, and from line263 to line 270.
5) It is difficult to understand the sentence from line 107 to line 109.
6) In line 146, “overexpression of SREBP-1 leads to atrophy with a decrease in muscle atrophic factors”, how to explain these results?
7) In line 198, “As mentioned above, a positive role of PKCδ is considered in insulin-stimulated glucose uptake in skeletal muscle cells. Muscle PKCδ levels increase with age in mice, and muscle-specific deletion of PKCδ improves muscle insulin resistance and whole-body insulin sensitivity in aged mice.”
For this part, PKCδ was not mentioned before, and from the description, it seems that PKCδ plays a negative role, but not positive role, in insulin-stimulated glucose uptake in skeletal muscle cells.
Overall, all the details demonstrate that the authors lack some basic knowledge for scientific writing or they didn’t pay necessary attention to the manuscript.
Author Response
Response to Reviewer
As reviewer inferring, skeletal muscle is responsible for the body’s energy expenditure, participating in glucose and lipid uptake, and other metabolic processes as well as thermogenic functions. We discussed mainly how excess FFAs and lipid accumulation in skeletal muscle impair insulin sensitivity and cellular homeostasis as well as informed the molecular mechanisms underlining altered insulin signaling pathways. And we tried to summarize the main mechanisms leading to the development of insulin resistance in skeletal muscle. However, increasing evidence supports that an association of declining skeletal muscle mass on insulin resistance has not been fully investigated, yet. We hope that this report may be informative to understand current consequences of metabolic responsibility of skeletal muscle. We really appreciate reviewer for bunch of comments and directions. We corrected reviewer’s direction as below and main manuscript, too.
Comments and Suggestions for Authors
The review manuscript entitled “Excess Accumulation of Lipid Impairs Insulin Sensitivity in Skeletal Muscle” introduces some recent progress in the area of insulin sensitivity in skeletal muscle. Lipid accumulation and insulin sensitivity are two important and hot topics in the study of metabolic disorders. New insights from these two topics will potentially benefit the treatment of diseases, such as type 2 diabetes.
Unfortunately, the manuscript has some critical issues.
1) From the title, some important questions are expected to be addressed:
How does excess accumulation of lipids develop in skeletal muscle? Where are the lipids from? What could be the molecular mechanisms behind this process? b) How accumulation of lipids changes insulin sensitivity and what could be the corresponding signaling? c) How could exercise or other interventions affect or restore insulin sensitivity targeting lipid accumulation, and what could be the mechanisms?For these important questions, the manuscript doesn’t provide clear and detailed answers or thoughts.
Answer; The mechanism(s) responsible for the accumulation of intramyocellular triacylglycerols and intermediates of lipid metabolism in intact skeletal muscle are not fully evident. Two possibilities include an increase in lipid synthesis or a reduction in fatty acid oxidation. Through the text, we have suggested theoretical evidence with several mechanistic explanations regarding the relationship between insulin signaling and glucose uptake leading to the accumulation of long-chain fatty acyl-CoAs in skeletal muscle.
2) The authors don’t provide full names for a lot of abbreviations. For example, it is hard to believe that the full names of GLUT4 and SREBP-1 are not included in the manuscript. This demonstrates that the authors paid little attention to the quality of the manuscript.
Answer; We corrected all main acronyms through text.
3) All the figures are not the summaries of corresponding parts. Figure 1 and 3 are general and simple figures without providing valuable information and summaries.
Answer; Figure 1 is to introduced to emphasize maintenance of energy homeostasis in skeletal muscle regarding insulin-stimulated glucose uptake occurs in skeletal muscle. Figure 3 shows clear effects of exercise-stimulated glucose uptake in skeletal muscle.
For figure 2, it is not clear how the authors came up with this figure, because there is no description or introduction for the interactions of different components in the figure in the manuscript. Besides, it is difficult to understand how insulin resistance develops from figure 2. Basically, the authors provide some results from current research, but cannot provide appropriate summaries and new thoughts or ideas.
Answer; We well describe the Fig 2 explaining the insulin signaling pathways and the mechanisms mediated by influx of lipid in the skeletal muscle via text chapter 3.
4) References are needed for the contents from line 240 to line 246, and from line263 to line 270.
Answer; Ref 65 and 66 regard the text from line 240 to line 246. Ref 72 and 73 are refering text from line263 to line 270.
5) It is difficult to understand the sentence from line 107 to line 109.
Answer; As corrected in reviewer 1 points, the sentence from line 107 to line 109 was changed to ‘ In skeletal muscle from type 2 diabetic patients, IRS-1 tyrosine phosphorylation, PI 3-kinase activity, and glucose transport activity were impaired, whereas insulin receptor tyrosine phosphorylation, MAP kinase phosphorylation, and glycogen synthase activity were normal [26, 27].’
6) In line 146, “overexpression of SREBP-1 leads to atrophy with a decrease in muscle atrophic factors”, how to explain these results?
Answer; In SREBP-1 overexpressing cells, a combined decrease in the expression of the myogenic regulatory factors MYOD1, MYOG and MEF2C previously was reported. The myogenic regulatory factors may be required to maintain muscle homeostasis, and may play a role in muscle plasticity in response to both hypertrophic (ie exercise) and atrophic (ie denervation) stimuli. Ref 42 was include regarding this comments. More information inferring that SREBP-1 leads to atrophy with a decrease in muscle atrophic factors is involved in Mol Cell Biol, 2010, 30: 1182–1198.
7) In line 198, “As mentioned above, a positive role of PKCδ is considered in insulin-stimulated glucose uptake in skeletal muscle cells. Muscle PKCδ levels increase with age in mice, and muscle-specific deletion of PKCδ improves muscle insulin resistance and whole-body insulin sensitivity in aged mice.”
For this part, PKCδ was not mentioned before, and from the description, it seems that PKCδ plays a negative role, but not positive role, in insulin-stimulated glucose uptake in skeletal muscle cells.
Overall, all the details demonstrate that the authors lack some basic knowledge for scientific writing or they didn’t pay necessary attention to the manuscript.
Answer; “As mentioned above “ was excluded. In young mice, muscle-specific deletion of PKCδ did not rescue high-fat diet-induced insulin resistance or glucose intolerance, however, with an increase in age, PKCδ levels in muscle increased, muscle-specific deletion of PKCδ improved whole-body insulin sensitivity and muscle insulin resistance and by 15 months of age improved the age-related decline in whole-body glucose tolerance. Reference # 53 was changed to Diabetes , 2015 Dec;64(12):4023-32.
Round 2
Reviewer 2 Report
The revised version only fixes some obvious mistakes, which should not happen, from the last version. The new version doesn’t address my concerns appropriately. For example, does figure 1 appropriately summarize “2. Insulin Signaling Pathway and Glucose Uptake in Skeletal Muscle.”? The answer is No. The same situation can be seen for figure 3.
Moreover, how does excess lipid accumulation develop in skeletal muscle? Under part 3. High Fat Accumulation Triggers Insulin Resistance in Skeletal Muscle, the authors only mentioned that when the amount of FFA exceeds the absorption capacity of the adipocytes, lipids spill over into non-adipose tissues, heart, liver, and skeletal muscle (line 131). Where are the mechanisms? Then they started to discuss the effect of glucose on SREBP-1, which actually is a different direction from the title of that part.